# Photocatalytic Degradation of Gaseous Benzene Using Cu/Fe-Doped TiO_2_ Nanocatalysts under Visible Light

**DOI:** 10.3390/molecules29010144

**Published:** 2023-12-26

**Authors:** Tao Tian, Jie Zhang, Lijiang Tian, Sijie Ge, Zhenyu Zhai

**Affiliations:** 1School of Environment Science and Spatial Informatics, China University of Mining and Technology, Xuzhou 221000, China; tt9807102023@163.com (T.T.); zhangjie743@126.com (J.Z.); gesijie@cumt.edu.cn (S.G.); 2Sinoma International Intelligent Technology Co., Nanjing 221122, China; zhenyu9807@163.com

**Keywords:** gaseous benzene, TiO_2_, Cu/Fe-doping, photocatalytic, visible light

## Abstract

Visible-light-enhanced TiO_2_ nanocatalysts doped with Cu and Fe were synthesized using the sol–gel method to investigate their performance in degrading gaseous benzene. The structure and morphology of mono- and co-doped TiO_2_ (i.e., Cu/Fe-TiO_2_, Cu-Fe-TiO_2_) were characterized using SEM, EDS, XRD, BET, Raman, UV-vis-DRS, and XPS techniques. The results indicated that the presence of Cu/Fe mono- and co-doped TiO_2_ leads to the formation of an anatase phase similar to pure TiO_2_. Furthermore, the introduction of Cu/Fe enhanced the presence of lattice defects and increased the specific surface area of TiO_2_. This enhancement can be attributed to the increase in oxygen vacancies, especially in the case of Cu-Fe-TiO_2_. Additionally, Cu-Fe-TiO_2_ showed a higher concentration of surface-bound hydroxyl groups/chemically adsorbed oxygen and a narrower bandgap than pure TiO_2_. Consequently, Cu-Fe-TiO_2_ exhibited the highest photocatalytic performance of 658.33 μgC_6_H_6_/(g·h), achieving a benzene degradation rate of 88.87%, surpassing that of pure TiO_2_ (5.09%), Cu-TiO_2_ (66.92%), and Fe-TiO_2_ (59.99%). Reusability tests demonstrated that Cu-Fe-TiO_2_ maintained a high benzene degradation efficiency of 71.4%, even after five experimental cycles, highlighting its exceptional stability and reusability. In summary, the addition of Cu/Fe to TiO_2_ enhances its ability to degrade gaseous benzene by prolonging the catalyst’s lifespan and expanding its photoresponse range to include visible light.

## 1. Introduction

Gaseous benzene is a typical volatile organic compound (VOC) that is widely generated in various industrial sectors, such as petroleum refining, chemical manufacturing, printing operations, and transportation [1]. It produces by-products that contribute to the formation of O_3_, which has detrimental effects on the environment, human health, and ecosystems. Several technologies have been employed for the degradation of gaseous benzene, including physical and chemical adsorption, biodegradation, plasma technology, membrane separation, and catalytic oxidation [2,3]. In contrast, semiconductor photocatalysis has gained significant attention in research due to its ability to effectively mineralize organic pollutants under mild reaction conditions. The key aspect of this technology lies in the development of efficient and stable photocatalysts [4]. TiO_2_ is widely used as a photocatalyst due to its favorable chemical properties, non-toxic nature, and resistance to photo-corrosion [5]. However, its photocatalytic efficiency is limited to UV light and does not respond to visible light [6,7]. Moreover, the performance of TiO_2_ is impeded by the recombination of electrons and holes generated during the photocatalytic process [8]. Therefore, current research focuses on reducing the band gap and electron–hole recombination rate of TiO_2_ to enhance its responsiveness to a wider spectrum of light and prolong the lifespan of photocatalysis. Recently, there has been significant interest in developing visible-light-induced photocatalysts using modified TiO_2_. These modified photocatalysts have shown promising results in effectively degrading pollutants [9,10]. One approach to achieve this is by establishing p-n-n heterojunctions or prolonging the lifespan of charges, thereby enhancing the structural properties of the photocatalysts. Consequently, enhancing the visible light response of TiO_2_-based materials with exceptional photocatalytic ability has become a key research focus in the field of photocatalysis [11].

To enhance the performance of TiO_2_ under visible light, researchers have focused on various technologies including ion doping, surface modification, and semiconductor coupling [12,13]. Among these, ion doping is considered the most effective method for altering the TiO_2_ surface. Researchers have investigated the use of non-metal-doped TiO_2_ to enhance its photocatalytic performance in degrading gaseous benzene. However, the improvements achieved through non-metal doping are not significant due to differences in valence state and ionic radius. For instance, the benzene degradation rate of N-doped TiO_2_ under visible light is only 24% [14]. On the other hand, incorporating the S element into the S-TiO_2_ system has shown promising results, with the photocatalytic degradation efficiency of benzene ranging from 34% to 41%, surpassing that of N-doped TiO_2_ [15]. Nevertheless, further improvements in catalytic efficiency are still required to meet practical demands.

Transition metal ion doping is a convenient and effective technology for modifying the surface of TiO_2_ [16,17]. This process involves overlapping the ns^2^ or empty d orbital of the metal ion with the 2p orbital of oxygen in TiO_2_, resulting in the creation of a new energy band within the wide bandgap of the semiconductor. Consequently, the band gap width of the semiconductor decreases [18]. Naghibiet found that Fe-doped TiO_2_ had the smallest crystal size and the narrowest forbidden band compared to Ce/Cd/Cu-doped TiO_2_, indicating that Fe-doped TiO_2_ could enhance photocatalytic performance [19]. Furthermore, Cu has shown potential as a photocatalyst for absorbing visible light [20,21,22]. When copper oxide is combined with TiO_2_, it exhibits excellent photocatalytic properties on various substrates or materials. The synergistic effect of this combination further enhances the photocatalytic activity [23,24]. This combination expands the range of light absorption to include the abundant visible wavelengths of solar radiation.

Unfortunately, mono-doping a metal into a crystal can disturb the distribution of charge within the crystal. This interference is primarily caused by changes in valence state and ionic radius, resulting in an increased recombination rate of photogenerated carriers and a decrease in the photocatalyst’s performance [25]. However, research has shown that incorporating different metal elements during doping can effectively regulate charge by introducing electron donors and acceptors. These elements interact with each other’s electronic structure without adversely affecting the charge density in the crystal lattice. This technique shows promise in minimizing the formation of unnecessary defects [26,27]. In recent years, researchers have extensively studied co-doping systems such as Cu-Co [28], Cu-Mo [29], and Cu-Mn [30] to enhance material properties. However, these co-doping systems have shown limited success. Hence, it is crucial to explore alternative co-doping systems that can improve the desired properties. Iron, with its multiple valence states, facilitates electron transfer between catalysts and reactants, thereby enhancing photochemical reactions [31]. Furthermore, doping TiO_2_ with Cu and Fe can results in distinct valence states, each possessing unique photocatalytic properties. However, there is still limited research on the doping of copper and iron in TiO_2_. Therefore, it holds great significance to synthesize Cu- and Fe-doped TiO_2_ photocatalysts and investigate their effectiveness in degrading gaseous benzene under visible light.

In this study, Cu/Fe mono- and co-doped TiO_2_ nanoparticles were synthesized using the sol–gel method. The synthesized catalysts were characterized using SEM, EDS, XRD, BET, Raman, UV-Vis, and XPS techniques to comprehensively analyze the effects of mono- and co-doping on the photocatalytic properties of TiO_2_. Furthermore, the photocatalytic performance of various catalysts was evaluated by measuring the degradation of gaseous benzene.

## 2. Experimental Part

### 2.1. Chemicals and Materials

Nitric acid (HNO_3_; AR) and Acetylacetone (C_5_H_8_O_2_; AR) were purchased from Sinopharm Chemical Reagent Company (Shanghai, China). Absolute ethanol (C_2_H_6_O; AR) was provided by SuYi Chemical Reagents (Shanghai, China). Copper nitrate (Cu(NO_3_)_2_•3H_2_O; AR), ferric nitrate (Fe(NO_3_)_3_•9H_2_O; AR), tetra-butyl titanate (C_16_H_36_O_4_Ti, CP, ≥98%), and benzene were purchased from Aladdin Biochemical Technology (Shanghai, China). All chemicals were used as received without further purification. 

### 2.2. Photocatalysis Experimental Setups

The photocatalytic reaction system consists of three main parts: gas transportation, reactor, and exhaust treatment (Figure 1). The gas delivery system includes a nitrogen gas purging system and an air pump to generate ambient air. Before introducing benzene and air into the reactor in the gas phase, they are thoroughly mixed using a gas mixing device. The reactor, made of quartz, is divided into three main parts: an optical condensing unit, a photocatalytic reactor, and a volatile organic compound tester (model RAE 3000) that measures concentrations in parts per billion (ppb). The light source condensing unit filters the UV rays emitted by the tungsten iodine lamps to ensure that only the desired wavelengths of UV light are used in the photocatalytic reaction. A condensation liquid containing a sodium nitrite solution with a concentration of 2 mol/L is employed to selectively utilize the specific wavelengths of UV light necessary for the photocatalytic reaction. Nickel foam is used as a catalyst carrier. To prepare the nickel foam, it is immersed in a 0.1 mol/L HNO_3_ solution for 15 min and then calcinated at 300 °C for 1 h. Subsequently, the nickel foam is pretreated in a 1:1 solution of titanate coupling agent and anhydrous ethanol for 15 min. Finally, the powdered catalyst is evenly applied onto the nickel foam. During testing, a black box is used to separate the reactor from outside sunlight. Additionally, activated carbon is used to adsorb any residual waste gas that remains undegraded during photocatalytic experiments.

### 2.3. Synthesis of Photocatalysts

Pure TiO_2_ was synthesized using the sol–gel method as previously reported [32]. Solution A was prepared by combining 25 mL of tetrabutyl titanate, 35 mL of ethanol, and 1.2 mL of acetylacetone. Similarly, solution B was prepared by combining 35 mL of ethanol, 1.2 mL of acetylacetone, and deionized water. Solution A was then gradually added to solution B at a rate of 1 drop/s while stirring. This process lasted 3 h to obtain a mixed solution. The mixed solution was then heated to 30 °C to promote the formation of a gel with high moisture content. Before gel formation, the pH value of the mixed solution was adjusted to 4.0 using nitric acid. The wet gel was subsequently centrifuged and dried at 100 °C to obtain dehydrated gel. Finally, the dried gel was calcined at 400 °C for 2 h to obtain pure TiO_2_.

Mono-doped TiO_2_ (i.e., Cu-TiO_2_, Fe-TiO_2_) was synthesized by adding a specific amount of Cu(NO_3_)_2_•3H_2_O or Fe(NO_3_)_3_•9H_2_O to liquid A. The remaining steps in the synthesis process are identical to those for pure TiO_2_. Co-doped TiO_2_ was synthesized by adding a certain amount of Cu(NO_3_)_2_•3H_2_O to liquid A and a specific quantity of Fe(NO_3_)_3_•9H_2_O to liquid B. The remaining steps in the synthesis process are also identical to those for pure TiO_2_.

### 2.4. Characterization Instrumentation

Scanning electron microscopy (SEM) (model: Zeiss Sigma 300+ Oxford Spectroscopy, Oberkochen, Germany) was used to examine the catalyst’s surface morphology and reduce its size from the micron level to the nanoscale. Energy Dispersive X-ray Spectroscopy (EDS) (model: Zeiss Sigma 300+ Oxford Spectroscopy, Oberkochen, Germany), and EDS Elemental Mapping was used to perform qualitative and quantitative analysis of elemental distributions. X-ray diffractometry (XRD) (model: D8 ADVANCE, Bruker, Germany) was used to analyze the crystalline phase composition and degree of crystallization of the catalysts. A confocal Raman spectrometer (model: Horiba scientific LabRAM, Osaka, Japan) and BET-specific surface area (model: Quantachrome NOVA 4200e, Rochester, NY, USA) were used to calculate specific surface areas and analyze micro- and macro-porosity. A UV-Vis diffuse reflection spectrometer was performed using a UV-2600 with ISR-2600 Plus (model: Shimazu, Japan) to estimate the catalyst’s optical properties. X-ray photoelectron spectroscopy (XPS) (model: */Escalab 250Xi, Waltham, MA, USA) with monochromatized aluminum anodic targets and an X-ray beam spot size of 650 μm was used to analyze the surface components and chemical status of the catalyst.

### 2.5. Assessment of Photocatalytic Benzene Degradation

The catalysts were quantitatively evaluated by measuring the degradation of benzene under photocatalytic conditions. The initial concentration of benzene was set at 90 ± 5 mg/m^3^, and the exposure time was 80 s. The experiment maintained a consistent relative humidity level of 40%. A catalyst mass of 5 g was used. The photocatalytic reaction was initiated by turning on the tungsten iodine light when the benzene concentrations in the gas phase before and after the reactor reached adsorption equilibrium. The tests were conducted for 80 min under each reaction condition until a steady state was reached. Real-time measurements of gaseous benzene concentrations were taken before and after the reactor. The benzene elimination (*t*) was calculated using the following formula:(1)$$\eta =\left(1-\frac{C_{t}}{C_{0}}\right)\times 100\%$$

The benzene degradation rate is represented by *η*, where *C*_0_ indicates the import concentration of gaseous benzene and *C_t_* indicates the export concentration. To assess the efficiency of the catalyst in the degradation process of benzene, we utilize the unit capacity metric. The unit capacity is calculated using the following formula:(2)$$\kappa =\frac{1000\cdot Q\cdot \int_{0}^{t}C_{t}-C_{0}dt}{mt}$$
where *κ* indicates the unit capacity, μg C_6_H_6_/(g·h); *Q* indicates the flow of total gas, (L/h); *m* indicates the mass of the catalyst (g).

## 3. Results and Discussion

### 3.1. SEM and EDS Analysis

SEM analysis was conducted to examine the morphology and structure of the pure and doped TiO_2_ catalysts. Figure 2a showed the morphology of the TiO_2_ catalyst, indicating that the TiO_2_ nanoparticles possess an irregular shape, with a few spherical particles dispersed throughout. These particles are evenly distributed on the surface, although some clustering can be observed. Figure 2b showed that the Fe-TiO_2_ nanoparticles are spherical in shape and uniformly distributed in an aggregated state. These nanoparticles partially overlap with the TiO_2_ catalyst. Additionally, the catalyst surface exhibited a multi-layer structure, which enhanced the penetration of gaseous benzene into the catalyst’s interior. This, in turn, increased the specific surface area and improved the efficiency of the reaction. Figure 2c showed the clear aggregation and uniform crystallization of Cu-TiO_2_, resulting in a dense and uniform surface. The aggregation phenomenon can be attributed to the hydrophilic nature of TiO_2_ nanoparticles, which causes them to aggregate through van der Waals forces. However, the introduction of Cu into TiO_2_ leads to significant aggregation, reducing the specific surface area and hindering the formation of active sites on the catalyst [33,34]. Figure 2d demonstrated the presence of numerous spherical nanoparticles on the surface of Cu-Fe-TiO_2_, indicating that the addition of Fe to Cu-TiO_2_ increases the specific surface area. This doping further enhances the catalyst’s efficiency in benzene degradation.

Element mapping of the Cu-Fe-TiO_2_ nanocatalyst is shown in Figure 2e. These images clearly demonstrate the uniform distribution of Fe, Cu, Ti, and O elements, indicating the absence of copper and iron particle accumulation. The EDS analysis of Cu-Fe-TiO_2_ in Figure 2f reveals the presence of Fe, Cu, Ti, and O elements. The quantitative results from EDS indicate that the mass percentages (W %) of Fe, Cu, Ti, and O are 0.69, 3.11, 45.51, and 50.69, respectively. The presence of Fe and Cu elements confirms the titanium shell doping and the infiltration of Fe and Cu into the TiO_2_ lattice.

### 3.2. XRD and BET Analysis

XRD analysis was conducted to investigate the crystal phase and structure of the synthesized catalyst powder, as shown in Figure 3. Results indicated that the pure, mono-, and co-doped TiO_2_ all exhibited a high-purity anatase phase, suggesting that the employed doping techniques did not cause a transformation to a tetragonal structure [35]. Furthermore, the catalyst did not exhibit the characteristic peaks associated with Cu/Fe, which could be attributed to their low concentration. Table 1 presented the anatase crystal size, lattice distortion, and specific surface area of the catalysts. The results indicated that both mono- and co-doped TiO_2_ have smaller particle sizes compared to pure TiO_2_, indicating that the presence of Cu or Fe ions hindered the crystallization process of TiO_2_. Among the four catalysts, the Cu-Fe-TiO_2_ catalyst exhibited the smallest crystal size of 16.4 nm. Different dopants cause varying lattice distortions within the catalyst structure. The introduction of a small amount of Fe^3+^ (less than 0.1% mole percentage) significantly deformed the crystal lattice and reduced the particle size. This could be attributed to the smaller ionic radius of Fe^3+^ (0.064 nm) compared to Cu^2+^ (0.073 nm) and Ti^4+^ (0.068 nm), suggesting that Fe^3+^ can replace Cu^2+^ in the TiO_2_ lattice structure. The BET analysis revealed that Cu-Fe-TiO_2_ had the highest specific surface area, approximately 140.71 m^2^/g. Therefore, co-doping of Cu and Fe effectively reduces particle size, enhances lattice distortion, and increases the specific surface area.

### 3.3. Raman Analysis

The Raman spectra of the synthesized catalysts is shown in Figure 4. The spectra of the pure, mono-, and co-doped catalysts exhibit five distinct bands associated with the anatase crystal phase. These bands are observed at approximately 147, 198, 398, 515, and 640 cm^−1^ and correspond to the E1g, E2g, B1g, A1g, and E3g inactivity modes, respectively [36]. No Cu or Fe peaks were detected in the Raman spectra, which aligns with the findings of the XRD analysis. Figure 4 also demonstrates a significant increase in the half-maximum width of the distinctive peaks after the doping procedure. This increase can be attributed to the reduction in particle size and alterations in specific lattice constants. Notably, the spectral band near 147 cm^−1^ of the doped catalyst exhibited changes, with the E1g peak of pure TiO_2_ distributed in different bands, namely 144.32 cm^−1^, 149.71 cm^−1^, 148.71 cm^−1^, and 150.99 cm^−1^, corresponding to Cu-TiO_2_, Fe-TiO_2_, and Cu-Fe-TiO_2_, respectively. In comparison to pure TiO_2_, a slight upward shift in the Venturi number of the E1g peak was observed, which can be attributed to oxygen vacancies, molecular distortions, and non-cooperative effects of molecular force fields [37].

### 3.4. UV-Vis-DRS Analysis

UV-Vis-DRS technology was utilized to examine the optical properties of the synthesized catalysts. Figure 5a shows the UV-visible spectra of the catalysts, indicating significant absorption across the UV to visible light range for all synthesized catalysts. The addition of copper and iron elements to the catalysts significantly broadened the absorption spectrum, thereby enhancing their ability to absorb visible light. Figure 5b was used to determine the band gap energy of the sample using the Kubelka–Munk formula. This involved plotting the relationship between (ahv)^2^ and photon energy (hv) [38]. The band gap (Eg) of pure TiO_2_ was evaluated to be 3.2 (±0.06) eV. However, our observations indicated that the doping process affected the band gap of the catalyst. The band gap energy of Cu-TiO_2_, Fe-TiO_2_, and Cu-Fe-TiO_2_ decreased from 3.27 eV to 2.97 eV, indicating that the addition of Cu and Fe to TiO_2_ enhances the stability of the band gap structure and improves the efficiency of visible light absorption. This finding suggests that these improvements have the potential to advance the photocatalytic mechanism [39].

### 3.5. XPS Analysis

XPS analysis was conducted to comprehensively examine the chemical composition of the catalysts. As shown in Figure 6, the catalysts surface mainly consists of Ti, C, and O. The presence of carbon in the XPS instrument can be attributed to the use of oily carbon for spectral calibration.

The high-resolution XPS spectrum of the Ti2p is shown in Figure 7a. The measured binding energies of Ti2p3/2 and Ti2p1/2 range from 457.47 eV to 458.57 eV and from 463.21 eV to 464.32 eV, respectively. These results indicate that Ti in Ti-O clusters predominantly exists in the +4 valence state. Furthermore, the valence state of Ti remains unchanged even in the presence of Cu and Fe ions. However, the Ti2p binding energy of the composite is slightly lower compared to pure TiO_2_. Specifically, the measured binding energy of Ti2p3/2 is 459.3 eV, while the binding energy of Ti2p1/2 is 465.0 eV. Consequently, the electron cloud density around titanium is reduced, affecting the binding energy of Ti2p orbitals in the TiO_2_ lattice defects and causing a decrease in binding energy.

The high-resolution XPS spectra of the O1s are shown in Figure 7b. Each sample exhibited two distinct peaks corresponding to O1s. The energy levels measured at 528.72 eV, 529.01 eV, and 529.86 eV indicated the presence of lattice oxygen in Fe-TiO_2_, Cu-Fe-TiO_2_, and Cu-TiO_2_, respectively. The increased binding energy in mono- and co-doped TiO_2_, compared to pure TiO_2_, suggests a partial substitution of the oxide state [40]. The Cu-Fe-TiO_2_ compound, resulting from the redox reaction between Cu and Fe, exhibited a higher binding energy than TiO_2_ doped with either Cu or Fe. The energy values for Cu-Fe-TiO_2_, Cu-TiO_2_, and Fe-TiO_2_ were measured as 531.03 eV, 531.97 eV, and 531.85 eV, respectively. These results indicated the presence of hydroxyl groups attached to the surface or chemisorbed oxygen in the Cu-Fe-TiO_2_, Cu-TiO_2_, and Fe-TiO_2_ systems. The experimental findings demonstrated that the Cu-Fe-TiO_2_ catalyst has a higher fraction of adsorbed oxygen compared to other catalysts.

The Cu2p scan of Cu-TiO_2_ and Cu-Fe-TiO_2_ is shown in Figure 7c. The binding energies of Cu-TiO_2_ at Cu2p1/2 and Cu2p3/2 were measured to be 951.99 eV and 932.34 eV, respectively. These values indicate the presence of both CuO and Cu_3_O_2_ states, suggesting complete and incomplete oxidation of Cu, respectively [41]. Only one passive band was observed in the spectrum of Cu-Fe-TiO_2_, indicating the reaction of Cu^2+^ with TiO_2_. However, it is important to note that XPS technology has a detection limit of 5 nm, which means that this reaction cannot be detected.

The Fe2p scan of Fe-TiO_2_ and Cu-Fe-TiO_2_ is shown in Figure 7d. The absence of a distinct peak shape suggests a low concentration of doped Fe^3+^. In contrast, Cu-Fe-TiO_2_ exhibited a clear dual structure in the iron-related region, with Fe2p1/2 and Fe2p3/2 peaks at binding energies of 722.72 eV and 710.32 eV, respectively. These observations indicate the presence of Fe^3+^. Interestingly, no iron oxide is detected in the Cu-Fe-TiO_2_ composite, possibly due to the redox interaction between Fe^3+^ and Cu^2+^ during the catalyst synthesis process. Furthermore, the decrease in the intensity of Fe2p peaks in both catalysts suggests that the addition of Fe^3+^ leads to a more uniform distribution within the TiO_2_ framework, rather than being concentrated solely on the surface layer of the nanospheres.

### 3.6. Catalytic Activity

The degradation curve of benzene catalysts under visible light is shown in Figure 8a. The results demonstrated that mono- and co-doped TiO_2_ catalysts are more effective in degrading benzene under visible light compared to pure TiO_2_. This can be attributed to the broadened light absorption spectrum, which enhances the photocatalytic performance. Among the four catalysts, Cu-Fe-TiO_2_ exhibited the highest rate of benzene degradation, achieving a remarkable rate of 88.87%. During photocatalysis, the presence of Cu in various valence states significantly reduces the recombination rate between photogenerated electrons and holes. Furthermore, the distribution of Fe on the titanium base helps reduce oxide formation, which can act as a recombination center for photogenerated carriers. The Cu-Fe-TiO_2_ catalyst surface has a higher concentration of hydroxyl groups or chemisorbed oxygen, thereby improving the efficiency of benzene degradation under visible light. Cu-TiO_2_, Fe-TiO_2_, and Cu-Fe-TiO_2_ were deactivated after approximately 18, 35, and 60 min, respectively. This decrease in the rate of benzene degradation can be attributed to the interaction between Cu^2+^ and Fe^3+^ in the mono- and co-doped systems. Specifically, Cu^2+^ transfers to the catalyst surface and reacts with oxygen to form copper oxide, which acts as a complex center. This leads to an increase in the number of electron–hole pairs at the available sites, thereby enhancing the complexing rate. However, the presence of reaction intermediates or by-products can deactivate or block the active sites on the catalyst surface, limiting the availability of active sites for the adsorption and degradation of benzene molecules and thus reducing the degradation rate [42]. Moreover, the reaction process can induce changes in the structure of Cu-Fe-TiO_2_, leading to the redistribution or migration of metal ions within the catalyst. This modification impacts the active sites and electronic properties of the catalyst, thereby affecting its capacity to adsorb and degrade benzene molecules. Furthermore, as the reaction progresses, the recombination rate between photogenerated electrons and holes could potentially rise, causing a decrease in the efficiency of the photocatalytic system and subsequently reducing the degradation rate.

The processing capability of the photocatalysts is illustrated in Figure 8b. Among the four tested photocatalysts, Cu-Fe-TiO_2_ demonstrated the highest processing capacity, reaching 658.33 μgC_6_H_6_/(g·h). By substituting Ti^4+^ with Cu^2+^ in the TiO_2_ crystal structure, lattice distortion and imperfections are introduced. This leads to the formation of an oxide layer on the catalyst surface, which traps electrons and hinders the generation of electron–hole pairs. On the other hand, Fe^3+^ can substitute TiO_2_ but is unable to capture electrons. However, when both Cu^2+^ and Fe^3+^ are incorporated into the catalyst, its valence state is altered. These metal ions efficiently capture electrons, preventing the formation of complexes with photogenerated carriers and ultimately enhancing the photocatalytic performance of the catalyst.

### 3.7. Reusability Tests

Gaseous benzene degradation tests require stable Cu-Fe-TiO_2_ photocatalysts for practical applications. To assess the stability of the photocatalyst, we conducted five consecutive degradation experiments. Prior to the photocatalytic degradation process, the Cu-Fe-TiO_2_ photocatalyst underwent centrifugation, purification, filtration, and drying. As shown in Figure 9, the results demonstrated that the photocatalytic degradation rates of Cu-Fe-TiO_2_ were 88.87%, 84.35%, 80.63%, 77.16%, and 71.74% for each consecutive experiment. Although there was a slight decrease in the degradation rate, the photocatalytic stability of the catalyst remained relatively high. This decrease can be attributed to the depletion of the material with repeated use and the partial deactivation of its surface photocatalytically active sites.

### 3.8. Reaction Mechanism

As shown in Figure 10, this study presents a mechanism to explain the enhanced photocatalytic effectiveness of Cu^2+^ and Fe^3+^ co−doped TiO_2_ catalysts in the degradation of gaseous benzene. Our findings indicate that the substitution of Ti^4+^ with Cu^2+^ and Fe^3+^ introduces a new energy level associated with impurities. This energy level is located between the valence band and the conduction band in the crystal structure of TiO_2_. Electrons from higher energy levels readily transfer to the conduction band, while electrons from the valence band can move to another impurity energy level introduced by the iron dopant. The addition of Cu^2+^/Fe^3+^ dopants to TiO_2_ photocatalysts reduces the band gap, expanding their absorption range in the visible light spectrum compared to pure TiO_2_. As a result, released electrons can migrate to the catalyst’s surface and react with adsorbed O_2_ and H_2_O molecules, ultimately leading to the degradation of gaseous benzene [43,44]. Furthermore, Cu^2+^ and Fe^3+^ can act as carrier traps, impeding carrier recombination and prolonging the time taken for carriers to reach the catalyst surface, thereby enhancing photocatalytic efficiency. During the photocatalytic degradation of gaseous benzene using TiO_2_, intermediates such as six-membered ring alcohols may form. The catalyst absorbs O_2_ and H_2_O from the surrounding air during the reaction. When activated by light sources, various free radicals like −O_2_- and −OH are generated. These radicals then react with water to produce H_2_O_2_ and C_6_H_6_(OH)_2_. Subsequently, H_2_O_2_ reacts with −O_2_- to form the −OH radical. Finally, C_6_H_6_(OH)_2_ interacts with various free radicals to produce CO_2_ and H_2_O.

## 4. Conclusions

In this study, mono- and co-doped TiO_2_ catalysts (i.e., Cu/Fe-TiO_2_ and Cu-Fe-TiO_2_) were synthesized using the sol–gel method. The degradation rate of gaseous benzene was investigated using Cu-TiO_2_, Fe-TiO_2_, and Cu-Fe-TiO_2_ as photocatalysts. The synthesized catalysts were characterized using various techniques including SEM, EDS, XRD, BET, Raman, UV-vis-DRS, and XPS. Results indicated that both mono- and co-doped TiO_2_ consisted mainly of the anatase phase. The introduction of Cu/Fe dopants into TiO_2_ resulted in a noticeable reduction in particle size and band gap, which led to an increase in lattice defects and specific surface area, particularly in co-doped TiO_2_. Cu-Fe-TiO_2_ demonstrated significant photocatalytic activity, achieving an efficiency of 88.87% in benzene degradation and a benzene treatment capacity of 658.33 μgC_6_H_6_/(g·h). This excellent performance can be attributed to the reduction in band gap resulting from the co-doping of Cu and Fe, which enhances its sensitivity to visible light. Additionally, the presence of Cu and Fe in different valence states on the titanium substrate effectively captures photogenerated electrons, improves the separation of electron–hole pairs, and extends their lifespan. Furthermore, the Cu-Fe-TiO_2_ catalyst demonstrated remarkable stability and reusability, even after five degradation cycles. Overall, this study highlights the potential of metal-doped TiO_2_ catalysts for effectively degrading gaseous benzene.

## Figures and Tables

**Figure 1 molecules-29-00144-f001:**
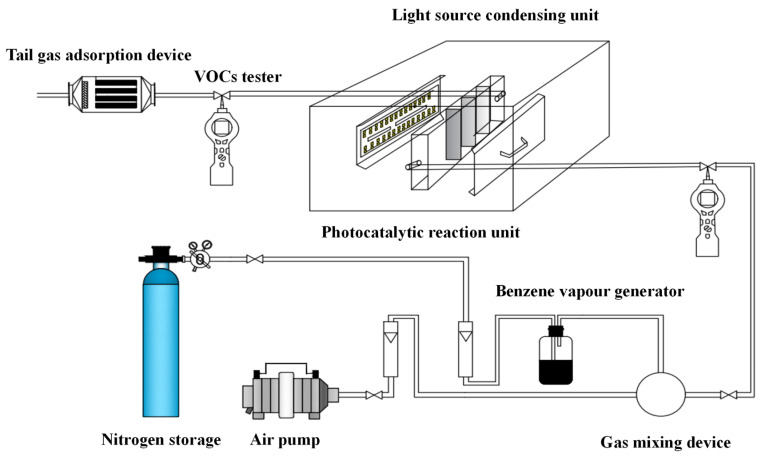
Flow diagram of photocatalytic experimental installation.

**Figure 2 molecules-29-00144-f002:**
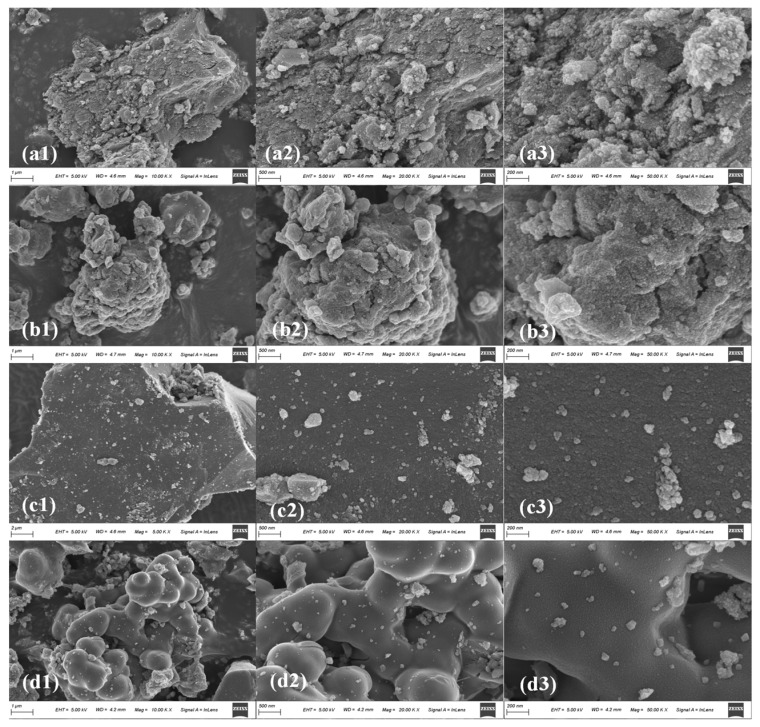
SEM micrographs of TiO_2_ catalysts under different loading metal: (**a1**–**a3**) TiO_2_, (**b1**–**b3**) Fe-TiO_2_, (**c1**–**c3**) Cu-TiO_2_, (**d1**–**d3**) Cu-Fe-TiO_2_, (**e**) elemental mapping of Cu-Fe-TiO_2_, EDS (**f**).

**Figure 3 molecules-29-00144-f003:**
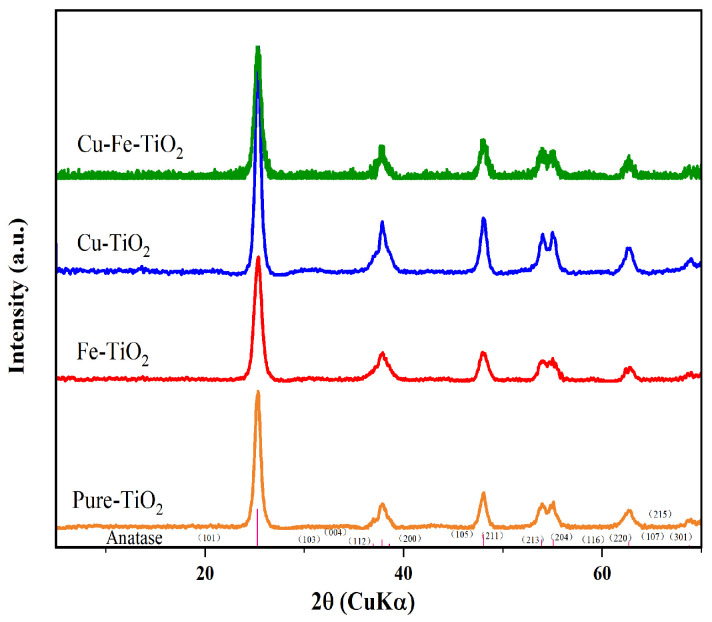
The X-ray diffraction patterns of the pure, mono-, and co-doped TiO_2_.

**Figure 4 molecules-29-00144-f004:**
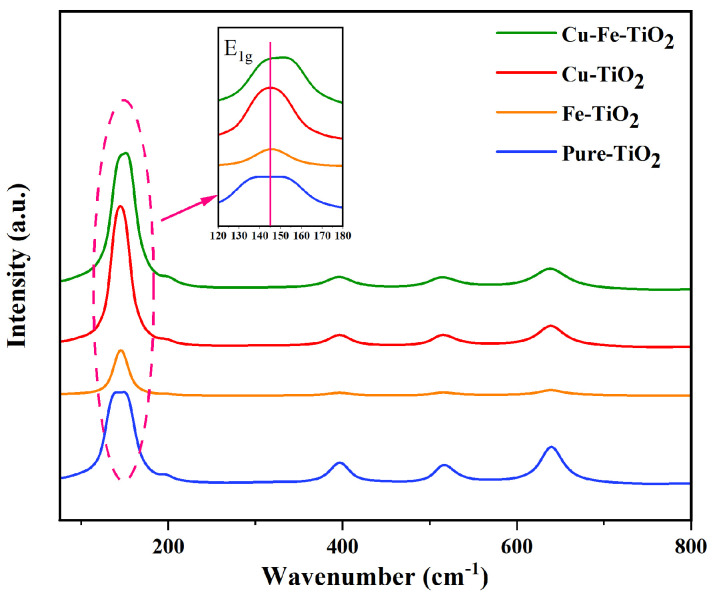
The Raman diffraction of the pure, mono−, and co−doped TiO_2_.

**Figure 5 molecules-29-00144-f005:**
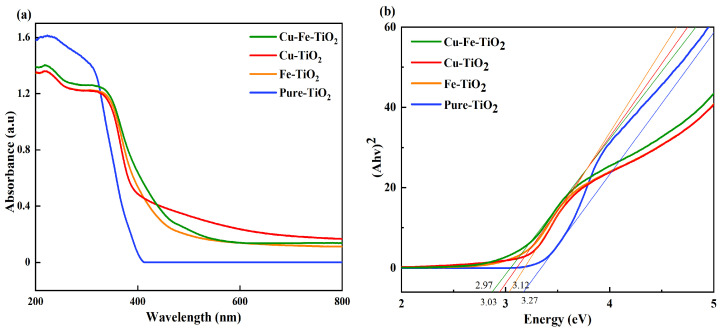
(**a**) UV-Vis diffuse reflection spectra (DRS); (**b**) plots of (ahv)^2^ vs. photon energy (hv) for the band gap energy.

**Figure 6 molecules-29-00144-f006:**
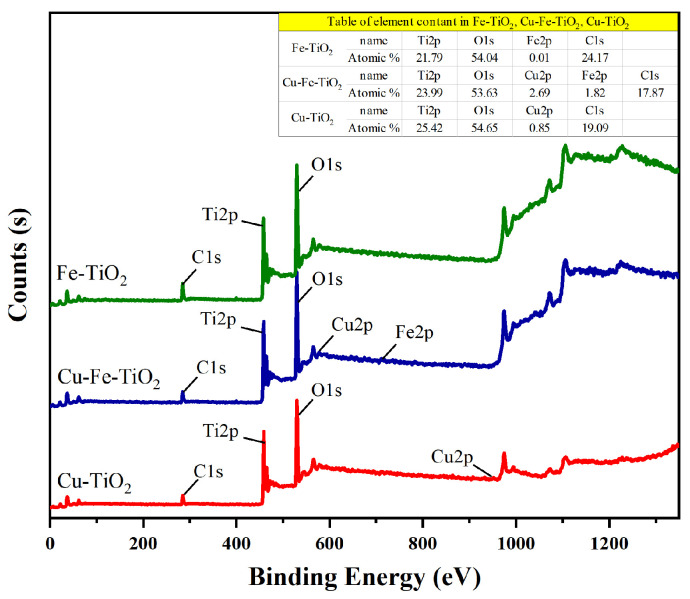
XPS survey spectrum of the pure, mono-, and co-doped TiO_2_.

**Figure 7 molecules-29-00144-f007:**
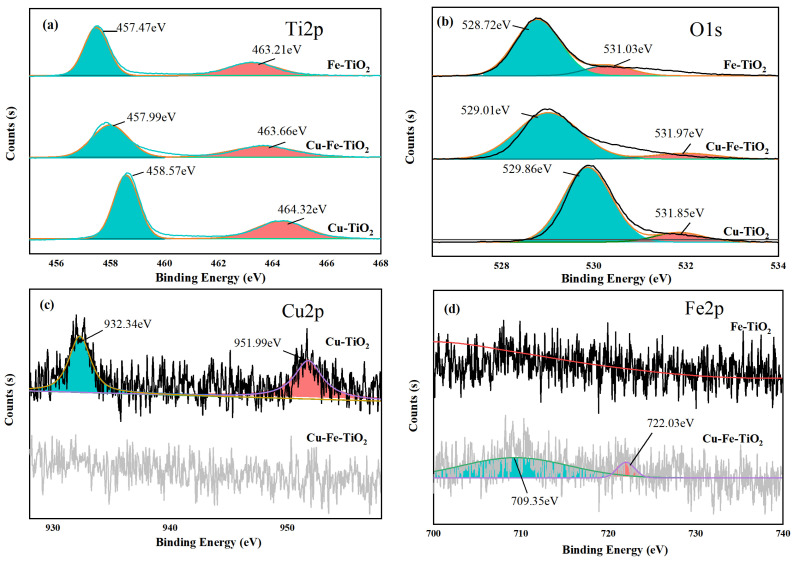
High-resolution XPS spectra: (**a**) Ti2p; (**b**) O1s; (**c**) I3d; (**d**) Fe2p.

**Figure 8 molecules-29-00144-f008:**
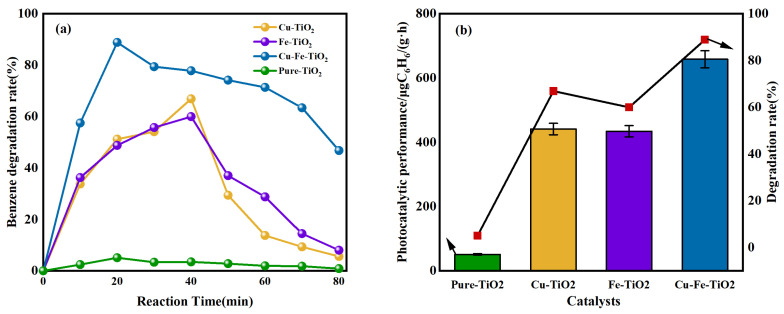
(**a**) The unit capacity of different catalysts to treat benzene under visible light; (**b**) photocatalytic performance of the obtained catalytic under visible light.

**Figure 9 molecules-29-00144-f009:**
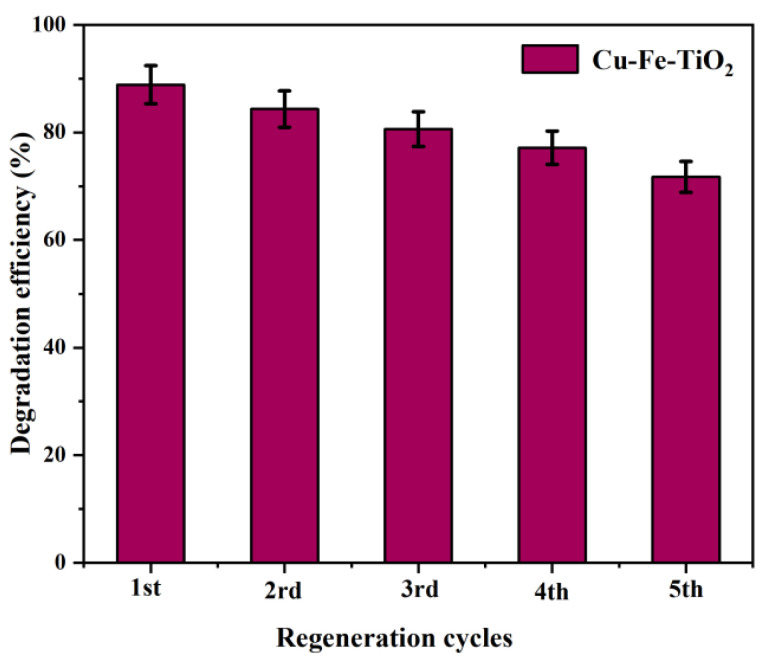
Cu-Fe-TiO_2_ photocatalyst stability experiments.

**Figure 10 molecules-29-00144-f010:**
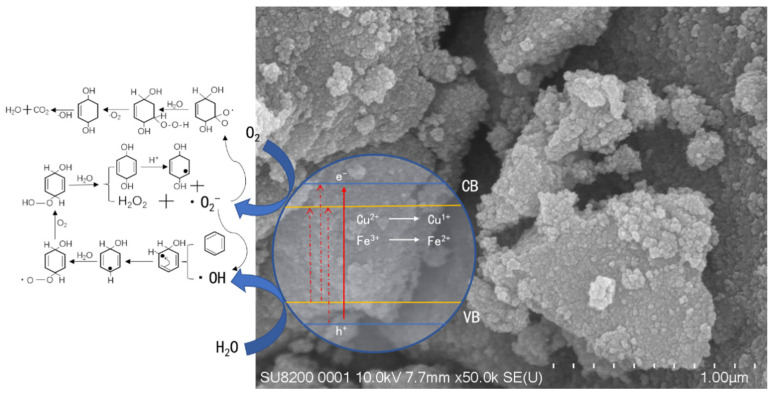
Photocatalytic degradation mechanism of benzene by Cu-Fe-TiO_2_.

**Table 1 molecules-29-00144-t001:** XRD crystallite size and BET surface area of the pure, mono-, and co-doped TiO_2_.

Sample	Pore Size (nm)	Lattice Distortion/ε	BET Surface Area (m^2^/g)
Pure-TiO_2_	43.0	0.246	32.73
Cu-TiO_2_	18.6	0.324	83.25
Fe-TiO_2_	20.9	0.358	62.47
Cu-Fe-TiO_2_	16.4	0.625	140.71

## Data Availability

All data generated or analyzed during this study are included in this published article.

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
