# Peer review of "Photocatalytic Degradation of Gaseous Benzene Using Cu/Fe-Doped TiO2 Nanocatalysts under Visible Light"

_molecules, 2023, doi:10.3390/molecules29010144_

Round 1
Reviewer 1 Report
Comments and Suggestions for Authors
The author reported the synthesis and characterization of metal-doped TiO2 and application in the photodegradation of gaseous benzene. The result is interesting and the prepared photocatalysts exhibits great potential in factory scale. However, the structural characterization of the hybrids and the discussion about the detailed photocatalytic mechanism is not enough in the present manuscript, so I recommend it to be published in Molecules after a major revision.
1. Where is the photocatalytic reactor? It should be shown in Figure 1 and the arrangement of the photocatalyst should be emphasized.
2. In Figure 2, difference in image (1) and (2) should be provided. SEM image of pure TiO2 should be provided.
3. Element mapping of the hybrids should be provided to observe the distribution of the metals.
4. How to demonstrate the doping of Fe in the Fe-TiO2 sample?
5. In Figure 8(a), how to understand the decline of the decomposition rate of the metal-doped TiO2 after certain time of reaction?
6. For the catalytic mechanism, authors are advised to provide more evidence, such as scavenging experiments, EPS measurements and so on.
7. Photocatalytic activity of photocatalysts are affected by many factors. Among them, band structure including the position of the conduction band and the valence band also determined the redox activity of the photocatalysts in addition to their band gaps. Especially the conduction band of the photocatalysts, which play a key role in determine the photocatalytic performance of catalysts, can be regulated by the doping of metal or non-metal doping process in recent reports (Chem. Eng. J. 2023, 466, 143219; J. Am. Chem. Soc. 2021, 143 (5), 2173; J. Colloid Interface Sci. 2023, 629, 336; Nano Lett. 2018, 18 (6), 3384). I wonder if the doping of metal ions affects the band structure of the hybrids?
8. The authors are advised to carefully read the whole manuscript to correct grammar and typos errors
Comments on the Quality of English LanguageThe authors are advised to carefully read the whole manuscript to correct grammar and typos errors
Reviewer 2 Report
Comments and Suggestions for Authors
The reviewed manuscript describes doping titania with copper and iron atoms. It was found that doped TiO2 is better photocatalyst than undoped TiO2.
The subject of the research is important for the air purification. The proposed photocatalysts can be used for the photocatalytic degradation of benzene in gaseous state.
The argumentation on structure of the catalysts is convincing. The obtained results agree with the literature on doped TiO2.
The cited references are relevant and are mostly recent publications. There are no excessive self-citations.
The manuscript is structured according to generally accepted standards. The conclusions are rather consistent with the technical background and the results obtained.
The sections 2.2, 2.5 and 3.3 of the manuscript are unclear. On the one hand, section 2.2 and Fig.1 suggest that the experiments were carried out in flow mode. On the other hand, the reusability tests in section 3.3 contradict the continuous flow mode. Section 2.5 informs that “light was deactivated to achieve adsorption equilibrium between benzene and the catalyst” (row 159). The provide reproducibility of the presented results, the experimental details should be clarified.
Please describe function of the light source condensing unit (optical condensing unit) mentioned in section 2.2. What is “the condensation liquid” (row 119) ?
Please describe the catalyst purification procedure mentioned in section 3.3.
Most of the figures are appropriate and clearly show the obtained results. However, some figures need to be improved:
Fig.1: Please check the direction of the air flow indicator. The phrase “nitrogen storage” is written twice.
Fig.8b: The mark and labels on the first vertical axis are inconsistent.
Fig.8b: Please indicate what data correspond to first and second axes.
The main problem is that photocatalytic activity changes over time. Fig.8a indicates clearly that benzene degradation rate is significantly decreased after 50 min of reaction. This fact should be explained.
Comments on the Quality of English LanguageThe manuscript contains multiple caps errors, for example:
Row 159: “A catalyst mass of 5 g was used. tungsten iodine light was deactivated to achieve adsorption equilibrium between benzene and the catalyst.”
Row 113: “Prior to introducing benzene and air into the reactor in the gas phase, Mix them thoroughly using a gas mixing device.”
Row 338: “Although the degradation rate decreased slightly, The photocatalytic stability of the catalyst remains relatively good.”
Row 63: “Nevertheless, Catalytic efficiency needs to be further improved to meet actual demand.”
Row 95: “In this study, We synthesized Cu/Fe mono and co doped TiO 2 via the sol gel technique.”
Please carefully check English. For example, “The high resolution XPS spectra of the O1s is shown in Fig 7(b).” (row 263).
Round 2
Reviewer 1 Report
Comments and Suggestions for Authors
It can be accept in the present form.
Reviewer 2 Report
Comments and Suggestions for Authors
The revised manuscript may be published